# Bioinformatics Analysis Identifying Key Biomarkers in Bladder Cancer

**Chuan Zhang** [ID]**, Mandy Berndt-Paetz and Jochen Neuhaus *** [ID]

Department of Urology, University of Leipzig, 04103 Leipzig, Germany;
Chuan.Zhang@medizin.uni-leipzig.de (C.Z.); Mandy.Berndt@medizin.uni-leipzig.de (M.B.-P.)
***** Correspondence: jochen.neuhaus@medizin.uni-leipzig.de; Tel.: +49-341-9717688

**Abstract:** Our goal was to find new diagnostic and prognostic biomarkers in bladder cancer (BCa), and to predict molecular mechanisms and processes involved in BCa development and progression. Notably, the data collection is an inevitable step and time-consuming work. Furthermore, identification of the complementary results and considerable literature retrieval were requested. Here, we provide detailed information of the used datasets, the study design, and on data mining. We analyzed differentially expressed genes (DEGs) in the different datasets and the most important hub genes were retrieved. We report on the meta-data information of the population, such as gender, race, tumor stage, and the expression levels of the hub genes. We include comprehensive information about the gene ontology (GO) enrichment analyses and the Kyoto Encyclopedia of Genes and Genomes (KEGG) pathway analyses. We also retrieved information about the up- and down-regulation of genes. All in all, the presented datasets can be used to evaluate potential biomarkers and to predict the performance of different preclinical biomarkers in BCa.

**Dataset:** The following are available online at http://www.mdpi.com/2306-5729/5/2/38/s1

**Dataset License:** CC-BY-NC

**Keywords:** bioinformatics; bladder cancer; biomarker; data

---

## 1. Summary

Bladder cancer is one of the most common malignancies [1]. Although new treatment strategies and tools for surgical resection [2], neoadjuvant chemotherapy [3,4], and photodynamic therapy (PDT) [5] have been developed, BCa remains with a high rate of recurrence [1]. Up to now, cystoscopy and bioptic histology are still the gold standards to diagnose bladder cancer (BCa) [6]. No consent about urinary marker or non-invasive screening strategies could be found by the European Association of Urology (EUA). Furthermore, nuclear matrix protein 22 (NMP22) is only recommended by American Urological Association (AUA) under certain conditions [7]. Therefore, it remains a priority to develop reliable, safe, and non-invasive diagnostic/prognostic biomarkers and therapeutic targets for BCa, and considerable efforts are ongoing.

Bioinformatics analysis is necessary for the integration of, e.g., huge amounts of transcriptome, microarray, and RNA-sequencing data to disclose alterations in gene expression, mutational burden, transcriptome, and proteome of cancer compared to non-cancer controls [8]. Of special importance are so-called hub genes, defined as highly connected genes, which can be regarded as a representative of a distinct module in the genes network [9]. Furthermore, hub genes potentially play an important role in the progression of cancer. Therefore, they are good biomarker candidates and may even provide new therapeutic targets [8,10]. Following, we provide supplemental results from our recent investigation

"Identification of key biomarkers in bladder cancer: Evidence from bioinformatics analysis" [11], which we share in a freely available and searchable form for further preclinical research. All in all, the data is the supplement for the previous research, providing selected data from the public The Cancer Genome Atlas (TCGA) and Gene Expression Omnibus (GEO) databases for cancer researchers and bioinformaticians to search and download for analysis based on the requisition of the public GDC Data Portal (https://portal.gdc.cancer.gov/).

## 2. Data Description

### 2.1. Database Analyses

Our previous study "Zhang et al. 2020, Identification of key biomarkers in bladder cancer: Evidence from bioinformatics analysis" [11] integrated six public datasets. Five datasets were downloaded from GEO (Gene Expression Omnibus, http://www.ncbi.nlm.nih.gov/geo) [12] and the bladder cancer (TCGA-BLCA) dataset was retrieved from TCGA (The Cancer Genome Atlas, https://portal.gdc.cancer.gov/) [13]. Below, we detailed the study design and the characteristics of this recent study.

We compared BCa samples and non-cancerous samples (histologically normal tissue adjacent to the tumor) and extracted six sets of differentially expressed genes (DEGs) from six public datasets according to the criteria for defining DEGs [11]. We used FUNRICH software [14] to map the DEGs and detected the overlap between the datasets (Table 1 and Table S1). We considered only DEGs, which were expressed in at least two of the five GEO databases (Table S2). Finally, we identified 418 DEGs (Table S3) overlapping with the 2537 DEGs from the TCGA-BLCA database (Table S3). Of those DEGs, 132 were upregulated and 286 were downregulated (Figure 1).

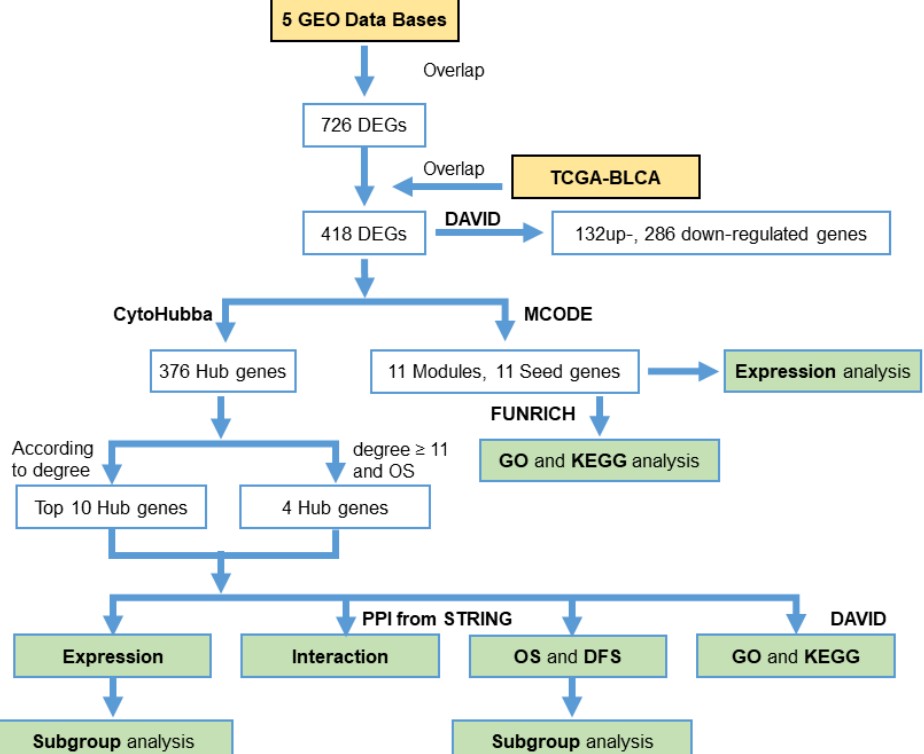

**Figure 1.** Study design and workflow of the analysis. Taken from Zhang et al. 2019 [11]. For detailed information on the analytical methods and the software packages please refer to the Materials and Methods section in the cited paper.

Molecular complex detection (MCODE) [15–17] analysis identified 11 relevant modules (subnetworks) and cytoHubba (based on Cytoscape software) [18] classified 376 of those 418 DEGs as hub genes, which are the gene most interconnected in the networks/modules (Table S4) [8].

To reduce the hub genes to the most promising, we defined 11 seed genes [15,16] for the most important 11 modules and the subsequent analysis yielded 14 hub genes on the basis of correlation to overall survival and degree of interaction (Table S5). The hub genes were ordered by their descending interaction degree: *CDK1*(98), *CCNB1*(92), *CCNA2*(84), *KIF11*(84), *CDC20*(83), *UBE2C*(83), *MAD2L1*(81), *AURKA*(80), *KIF20A*(80), *KIF2C*(80), *KPNA2*(67), *TPM1*(29), *CASQ2*(11), and *CRYAB*(11). Figure 2 depicts the results of the protein–protein interaction (PPI) analysis.

**Table 1.** General information on Gene Expression Omnibus (GEO) and The Cancer Genome Atlas (TCGA) datasets (adapted from: Zhang et al. 2020 [11]).

| Dataset | Number of Noncancerous Bladder Tissue Samples | Number of Cancer Tissue Samples | Number of DEGs Extracted from Dataset | Number of DEGs after FUNRICH Mapping |
|---|---|---|---|---|
| GSE27448 [19,20] | 5 | 10 | 5251 | 4701 |
| GSE52519 [21] | 3 | 9 | 751 | 742 |
| GSE61615 [22] | 2 | 2 | 842 | 736 |
| GSE76211 [23,24] | 3 | 3 | 770 | 658 |
| GSE100926 [25] | 3 | 3 | 223 | 194 |
| TCGA-BLCA [13] | 19 | 406 | 2873 | 2537 |

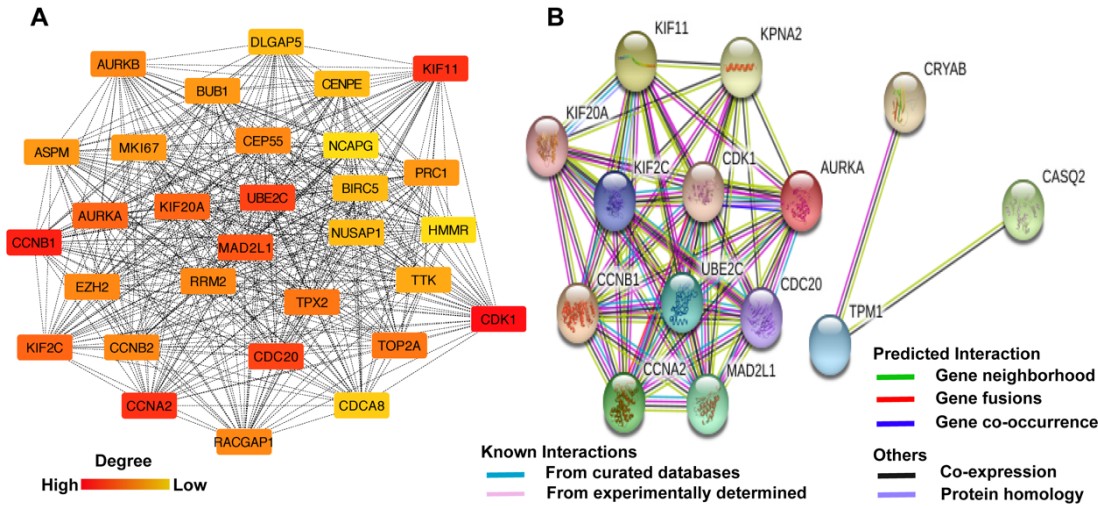

**Figure 2.** Interaction network analysis. (**A**) The interactions and protein–protein networks of the top 30 hub genes. (**B**) Protein-protein network and interaction among the 14 hub genes from STRING-db.org, accessed on 11 November 2019. Nodes with different colors represent different query proteins. A different color on the edge means a different interaction (see legend in figure). Adapted from Figure 4 of Zhang et al. 2020 [11].

We then performed a gene ontology (GO) enrichment analysis [26] and used the Kyoto Encyclopedia of Genes and Genomes (KEGG) pathway analysis [27] to identify the pathways potentially related to these protein-coding genes and to predict the roles of these genes in BCa. The Go and KEGG analyses significantly enriched the 'Pathways in cancers', 'Viral carcinogenesis', and 'Cell cycle', which relate to carcinogenesis and progression of cancer. We listed all significant results of GO and KEGG analysis based on the 418 DEGs and Benjamini–Hochberg value <0.05 in Table S6. The GO and KEGG analysis results based on the 14 hub genes are available and searchable at http://www.mdpi.com/2075-4418/10/2/66/s1, Table S5. Moreover, we also listed all significant results of GO and KEGG analyses of up- (Table S7) and downregulated DEGs (Table S8). In addition, we

extracted the clinical meta-data from TCGA-BLCA for the correlation to overall survival (OS) and disease-free survival (DFS) based on the 14 hub genes (Table S5).

In addition, we performed other subgroup analyses, such as the expression levels in different groups based on tumor stage, lymph nodal metastasis, race of patients, gender of patients, histological subtype, and molecular subtypes. We found that *CKD1, CCNB1, CCNA2, KIF11, CDC20, UBE2C, MD2L1, AURKA, KIF20A, KIF2C, KPNA2, TPM1, CASQ2,* and *CRYAB* were significantly higher expression in Caucasian and African American than in the ASI cohort. Except for TPM1, CASQ2, and CRYAB, all the genes were significantly overexpressed in both, male and female bladder cancer patients. However, no significant difference was found between males and females. All the genes were significantly higher expressed in non-papillary tumors than in papillary tumors (Table 2) [28]. In addition, *CKD1, CCNB1, CCNA2, KIF11, CDC20, UBE2C, MD2L1, AURKA, KIF20A, KIF2C,* and *KPNA2* were significantly upregulated in papillary tumors and non-papillary tumors than in non-cancerous tissues; in contrast, *TPM1, CASQ2,* and *CRYAB* were significantly downregulated in papillary tumors and non-papillary tumors than in non-cancerous tissues. Intriguingly, except for *CRYAB*, we found that TPM1 and *CASQ2* were most significantly downregulated in "Luminal Papillary" tumors, while the other genes were most significantly upregulated in subtype of "Neuronal" and "Basal squamous" based on molecular subtyping (Table 2).

## 2.2. Literature Research

We retrieved seven bioinformatics studies for BCa biomarkers based on public database analyses [9,29–34], and we compared the biomarkers in the present study with the ones that were reported in the literature studies. We found that *CRYAB* and *CASQ2* were so far unrecognized as biomarkers in previous studies. On the basis of Oncomine meta-analysis (https://www.oncomine.org/), we here present the meta-analysis of the expression levels of the hub genes described, but not been shown in our previous study (Figure 3 and Table S9). The genes compared in the meta-analysis were *CCNB1, CCNA2, KIF11, CDC20, UBE2C, MAD2L1, AURKA, KIF2C, CASQ2, CRYAB,* and *KIF20A*. Furthermore, except for the *CRYAB* and *CASQ2*, which have been shown in our previous paper, we constructed the expression body maps of the other 12 hub genes reported in our previous study using GEPIA (http://gepia.cancer-pku.cn, accessed on Nov.11. 2019). Body maps are an impressive way to visualize the differences in gene expression between normal and tumor tissues (Figure 4). Ultimately, our research results were roughly in line with the majority of the retrieved studies.

Table 2. Statistically significant difference of expression levels of target genes in subgroups.

| Target Genes | Race of Patients | | Gender of Patients | | Histological Subtypes | | Molebular Subtypes | |
|---|---|---|---|---|---|---|---|---|
| CDK1 | CAU (↑) vs. ASI | $p = 8.829 \times 10^{-4}$ | N vs. M (↑) | $p < 1.000 \times 10^{-12}$ | N vs. PT (↑) | $p = 2.109 \times 10^{-15}$ | N vs. NET (↑) | $p = 5.922 \times 10^{-8}$ |
| | AFA (↑) vs. ASI | $p = 3.379 \times 10^{-3}$ | N vs. F (↑) | $p = 1.554 \times 10^{-15}$ | N vs. NPT (↑) | $p = 1.624 \times 10^{-12}$ | N vs. BST (↑) | $p = 1.624 \times 10^{-12}$ |
| | | | | | PT vs. NPT (↑) | $p = 3.872 \times 10^{-2}$ | N vs. LT (↑) | $p = 5.809 \times 10^{-7}$ |
| | | | | | | | N vs. LIT (↑) | $p = 2.824 \times 10^{-12}$ |
| | | | | | | | N vs. LPT (↑) | $p = 1.625 \times 10^{-12}$ |
| CCNB1 | CAU (↑) vs. ASI | $p = 7.479 \times 10^{-7}$ | N vs. M (↑) | $p = 5.311 \times 10^{-13}$ | N vs. PT (↑) | $p = 2.949 \times 10^{-10}$ | N vs. NET (↑) | $p = 1.620 \times 10^{-5}$ |
| | AFA (↑) vs. ASI | $p = 5.221 \times 10^{-4}$ | N vs. F (↑) | $p = 2.907 \times 10^{-12}$ | N vs. NPT (↑) | $p = 2.631 \times 10^{-14}$ | N vs. BST (↑) | $p = 1.624 \times 10^{-12}$ |
| | | | | | PT vs. NPT (↑) | $p = 2.944 \times 10^{-3}$ | N vs. LT (↑) | $p = 2.202 \times 10^{-4}$ |
| | | | | | | | N vs. LIT (↑) | $p = 4.952 \times 10^{-7}$ |
| | | | | | | | N vs. LPT (↑) | $p = 3.502 \times 10^{-9}$ |
| CCNA2 | CAU (↑) vs. ASI | $p = 5.447 \times 10^{-8}$ | N vs. M (↑) | $p = 5.311 \times 10^{-9}$ | N vs. PT (↑) | $p = 4.309 \times 10^{-7}$ | N vs. NET (↑) | $p = 5.481 \times 10^{-7}$ |
| | AFA (↑) vs. ASI | $p = 9.961 \times 10^{-4}$ | N vs. F (↑) | $p = 3.062 \times 10^{-8}$ | N vs. NPT (↑) | $p = 4.395 \times 10^{-10}$ | N vs. BST (↑) | $p = 1.863 \times 10^{-12}$ |
| | | | | | PT vs. NPT (↑) | $p = 6.169 \times 10^{-4}$ | N vs. LT (↑) | $p = 2.176 \times 10^{-4}$ |
| | | | | | | | N vs. LIT (↑) | $p = 6.473 \times 10^{-5}$ |
| | | | | | | | N vs. LPT (↑) | $p = 4.175 \times 10^{-6}$ |
| KIF11 | CAU (↑) vs. ASI | $p = 5.620 \times 10^{-7}$ | N vs. M (↑) | $p = 5.836 \times 10^{-9}$ | N vs. PT (↑) | $p = 8.989 \times 10^{-8}$ | N vs. NET (↑) | $p = 4.693 \times 10^{-7}$ |
| | AFA (↑) vs. ASI | $p = 1.065 \times 10^{-6}$ | N vs. F (↑) | $p = 1.036 \times 10^{-7}$ | N vs. NPT (↑) | $p = 9.950 \times 10^{-10}$ | N vs. BST (↑) | $p = 2.290 \times 10^{-13}$ |
| | | | | | PT vs. NPT (↑) | $p = 6.997 \times 10^{-3}$ | N vs. LT (↑) | $p = 1.922 \times 10^{-5}$ |
| | | | | | | | N vs. LIT (↑) | $p = 4.399 \times 10^{-5}$ |
| | | | | | | | N vs. LPT (↑) | $p = 5.848 \times 10^{-7}$ |
| CDC20 | CAU (↑) vs. ASI | $p = 5.038 \times 10^{-3}$ | N vs. M (↑) | $p < 1.000 \times 10^{-12}$ | N vs. PT (↑) | $p = 1.691 \times 10^{-12}$ | N vs. NET | $p = 2.644 \times 10^{-8}$ |
| | AFA vs. ASI | $p = 3.077 \times 10^{-3}$ | N vs. F (↑) | $p < 1.000 \times 10^{-12}$ | N vs. NPT (↑) | $p < 1.000 \times 10^{-12}$ | N vs. BST | $p < 1.000 \times 10^{-12}$ |
| | | | | | PT vs. NPT (↑) | $p = 9.644 \times 10^{-5}$ | N vs. LT | $p = 8.558 \times 10^{-8}$ |
| | | | | | | | N vs. LIT | $p = 2.198 \times 10^{-11}$ |
| | | | | | | | N vs. LPT | $p = 3.194 \times 10^{-14}$ |
| UBE2C | CAU (↑) vs. ASI | $p = 6.099 \times 10^{-3}$ | N vs. M (↑) | $p < 1.000 \times 10^{-12}$ | N vs. PT (↑) | $p = 1.624 \times 10^{-12}$ | N vs. NET (↑) | $p = 1.045 \times 10^{-8}$ |
| | | | N vs. F (↑) | $p < 1.000 \times 10^{-12}$ | N vs. NPT (↑) | $p = 1.624 \times 10^{-12}$ | N vs. BST (↑) | $p = 1.624 \times 10^{-12}$ |
| | | | | | PT vs. NPT (↑) | $p = 1.429 \times 10^{-2}$ | N vs. LT (↑) | $p = 1.664 \times 10^{-10}$ |
| | | | | | | | N vs. LIT (↑) | $p = 1.497 \times 10^{-9}$ |
| | | | | | | | N vs. LPT (↑) | $p = 1.624 \times 10^{-12}$ |
| MAD2L1 | CAU (↑) vs. ASI | $p = 1.627 \times 10^{-7}$ | N vs. M (↑) | $p = 4.241 \times 10^{-14}$ | N vs. PT (↑) | $p = 1.634 \times 10^{-10}$ | N vs. NET (↑) | $p = 4.815 \times 10^{-7}$ |
| | AFA (↑) vs. ASI | $p = 4.381 \times 10^{-4}$ | N vs. F (↑) | $p = 1.488 \times 10^{-12}$ | N vs. NPT (↑) | $p = 1.625 \times 10^{-12}$ | N vs. BST (↑) | $p < 1.000 \times 10^{-12}$ |
| | | | | | PT vs. NPT (↑) | $p = 3.153 \times 10^{-4}$ | N vs. LT (↑) | $p = 1.755 \times 10^{-6}$ |
| | | | | | | | N vs. LIT (↑) | $p = 1.364 \times 10^{-7}$ |
| | | | | | | | N vs. LPT (↑) | $p = 1.625 \times 10^{-10}$ |

Table 2. *Cont.*

| Target Genes | Race of Patients | | Gender of Patients | | Histological Subtypes | | Molecular Subtypes | |
|---|---|---|---|---|---|---|---|---|
| *AURKA* | CAU (↑) vs. ASI | $p = 1.565 \times 10^{-7}$ | N vs. M (↑) | $p < 1.000 \times 10^{-12}$ | N vs. PT (↑) | $p = 6.550 \times 10^{-15}$ | N vs. NET (↑) | $p = 1.539 \times 10^{-7}$ |
| | AFA (↑) vs. ASI | $p = 1.629 \times 10^{-4}$ | N vs. F (↑) | $p < 1.000 \times 10^{-12}$ | N vs. NPT (↑) | $p < 1.000 \times 10^{-12}$ | N vs. BST (↑) | $p < 1.000 \times 10^{-12}$ |
| | | | | | PT vs. NPT (↑) | $p = 2.996 \times 10^{-3}$ | N vs. LT (↑) | $p = 2.259 \times 10^{-10}$ |
| | | | | | | | N vs. LIT (↑) | $p = 1.487 \times 10^{-11}$ |
| | | | | | | | N vs. LPT (↑) | $p = 1.674 \times 10^{-12}$ |
| *KIF20A* | CAU (↑) vs. ASI | $p = 6.312 \times 10^{-7}$ | N vs. M (↑) | $p = 4.398 \times 10^{-8}$ | N vs. PT (↑) | $p = 2.500 \times 10^{-7}$ | N vs. NET (↑) | $p = 4.343 \times 10^{-7}$ |
| | AFA (↑) vs. ASI | $p = 1.179 \times 10^{-4}$ | N vs. F (↑) | $p = 1.725 \times 10^{-6}$ | N vs. NPT (↑) | $p = 7.345 \times 10^{-9}$ | N vs. BST (↑) | $p = 1.373 \times 10^{-9}$ |
| | | | | | | | N vs. LT (↑) | $p = 7.949 \times 10^{-5}$ |
| | | | | | | | N vs. LIT (↑) | $p = 8.399 \times 10^{-4}$ |
| | | | | | | | N vs. LPT (↑) | $p = 7.846 \times 10^{-7}$ |
| *KIF2C* | CAU (↑) vs. ASI | $p = 1.844 \times 10^{-5}$ | N vs. M (↑) | $p = 1.624 \times 10^{-12}$ | N vs. PT (↑) | $p = 7.327 \times 10^{-15}$ | N vs. NET (↑) | $p = 1.387 \times 10^{-7}$ |
| | AFA (↑) vs. ASI | $p = 1.828 \times 10^{-5}$ | N vs. F (↑) | $p < 1.000 \times 10^{-12}$ | N vs. NPT (↑) | $p = 1.624 \times 10^{-12}$ | N vs. BST (↑) | $p = 1.624 \times 10^{-12}$ |
| | | | | | PT vs. NPT (↑) | $p = 2.143 \times 10^{-3}$ | N vs. LT (↑) | $p = 1.563 \times 10^{-10}$ |
| | | | | | | | N vs. LIT (↑) | $p = 4.494 \times 10^{-13}$ |
| | | | | | | | N vs. LPT (↑) | $p = 1.625 \times 10^{-12}$ |
| *KPNA2* | CAU (↑) vs. ASI | $p = 2.028 \times 10^{-8}$ | N vs. M (↑) | $p = 1.364 \times 10^{-11}$ | N vs. PT (↑) | $p = 2.543 \times 10^{-9}$ | N vs. NET (↑) | $p = 1.269 \times 10^{-8}$ |
| | AFA (↑) vs. ASI | $p = 1.019 \times 10^{-4}$ | N vs. F (↑) | $p = 2.920 \times 10^{-11}$ | N vs. NPT (↑) | $p = 2.059 \times 10^{-12}$ | N vs. BST (↑) | $p = 1.626 \times 10^{-12}$ |
| | | | | | PT vs. NPT (↑) | $p = 3.995 \times 10^{-4}$ | N vs. LT (↑) | $p = 6.761 \times 10^{-7}$ |
| | | | | | | | N vs. LIT (↑) | $p = 9.536 \times 10^{-8}$ |
| | | | | | | | N vs. LPT (↑) | $p = 5.829 \times 10^{-8}$ |
| *TPM1* | CAU (↑) vs. ASI | $p = 2.199 \times 10^{-11}$ | N vs. M (↓) | $p = 3.825 \times 10^{-3}$ | N vs. PT (↓) | $p = 3.180 \times 10^{-3}$ | N vs. NET (↓) | $p = 3.638 \times 10^{-3}$ |
| | AFA (↑) vs. ASI | $p = 7.136 \times 10^{-3}$ | N vs. F (↓) | $p = 4.022 \times 10^{-3}$ | N vs. NPT (↓) | $p = 4.757 \times 10^{-3}$ | N vs. BST (↓) | $p = 4.404 \times 10^{-3}$ |
| | | | | | PT vs. NPT (↑) | $p = 1.321 \times 10^{-6}$ | N vs. LT (↓) | $p = 3.798 \times 10^{-3}$ |
| | | | | | | | N vs. LIT (↓) | $p = 7.343 \times 10^{-3}$ |
| | | | | | | | N vs. LPT (↓) | $p = 2.404 \times 10^{-3}$ |
| *CASQ2* | CAU (↑) vs. ASI | $p = 2.406 \times 10^{-7}$ | N vs. M (↓) | $p = 4.142 \times 10^{-3}$ | N vs. PT (↓) | $p = 3.699 \times 10^{-3}$ | N vs. NET (↓) | $p = 3.638 \times 10^{-3}$ |
| | AFA (↑) vs. ASI | $p = 4.344 \times 10^{-2}$ | N vs. F (↓) | $p = 3.979 \times 10^{-3}$ | N vs. NPT (↓) | $p = 4.741 \times 10^{-3}$ | N vs. BST (↓) | $p = 4.404 \times 10^{-3}$ |
| | CAU vs. AFA (↑) | $p = 2.058 \times 10^{-2}$ | | | PT vs. NPT (↑) | $p = 1.379 \times 10^{-3}$ | N vs. LT (↓) | $p = 3.798 \times 10^{-3}$ |
| | | | | | | | N vs. LIT (↓) | $p = 7.343 \times 10^{-3}$ |
| | | | | | | | N vs. LPT (↓) | $p = 2.404 \times 10^{-3}$ |
| *CRYAB* | CAU (↑) vs. ASI | $p = 4.492 \times 10^{-8}$ | | | PT vs. NPT (↑) | $p = 4.366 \times 10^{-4}$ | | |
| | AFA (↑) vs. ASI | $p = 1.972 \times 10^{-2}$ | | | | | | |

The findings with *p*-value < 0.05 (Benjamini–Hochberg) were shown above. CAU (Caucasian), AFA (African American); ASI (Asian); male (M), $n = 297$; female (F), $n = 105$; Normal (N), $n = 19$; papillary tumors (PT), $n = 132$; non-papillary tumors (NPT), $n = 271$; molecular subgroups: neuronal tumors (NET), $n = 20$; basal squamous tumors (BST, $n = 142$; luminal tumors (LT), $n = 26$; luminal-infiltrated tumors (LIT): $n = 78$; luminal-papillary tumors (LPT) $n = 142$. *p* (*p*-value).

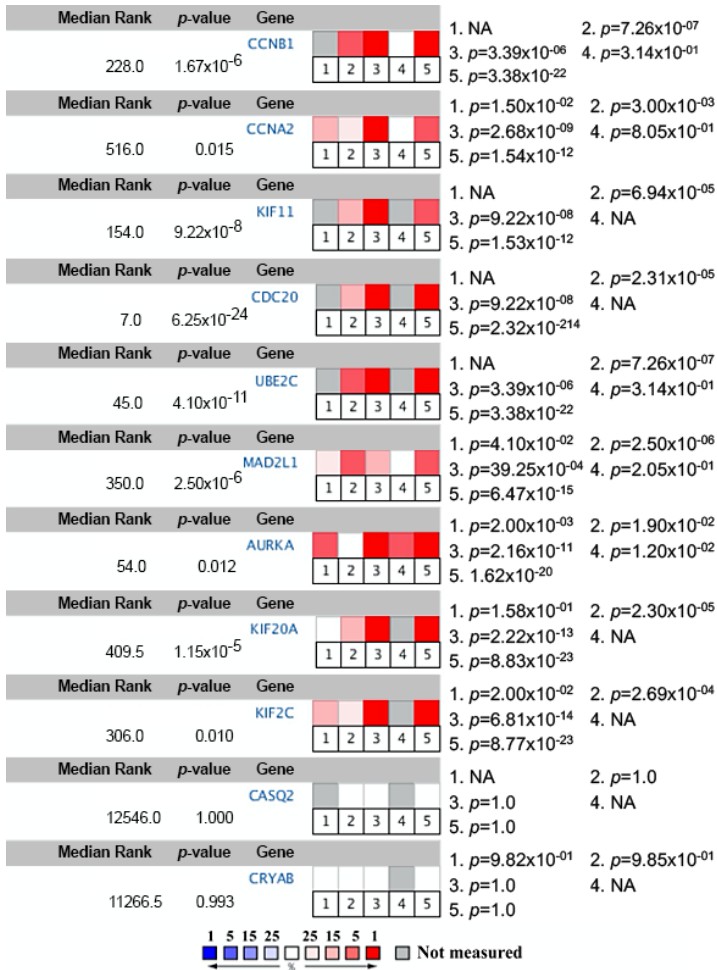

**Figure 3.** Oncomine meta-analysis of Hub genes in bladder cancer (BCa) vs. non-cancerous tissue. Left hand: median rank (median rank of the gene across each of the analyses); *p*-value (*p*-value for the median-ranked analysis); color of the boxes indicate the percentile of the z-transformed expression level of the gene in the particular study; right hand: *p* = (*p*-value reported in each of the studies); NA (not measured in the study); (1) Blaveri et al., Clin Cancer Res, 2005 [35], invasive cancer samples *n* = 51, normal bladder samples *n* = 3; (2) Dyrskjot et al., Cancer Res, 2004 [36], invasive cancer samples *n* = 13, normal bladder samples *n* = 14; (3) Lee et al., J Clin Oncol, 2010 [37], invasive cancer samples *n* = 62, normal bladder samples *n* = 10; (4) Modlich et al. Clin Cancer Res, 2004 [38], invasive cancer samples *n* = 20, normal bladder samples *n* = 4; and (5). Sanchez-Carbayo et al., J Clin Oncol, 2006 [39], invasive cancer samples *n* = 72, normal bladder samples *n* = 52.

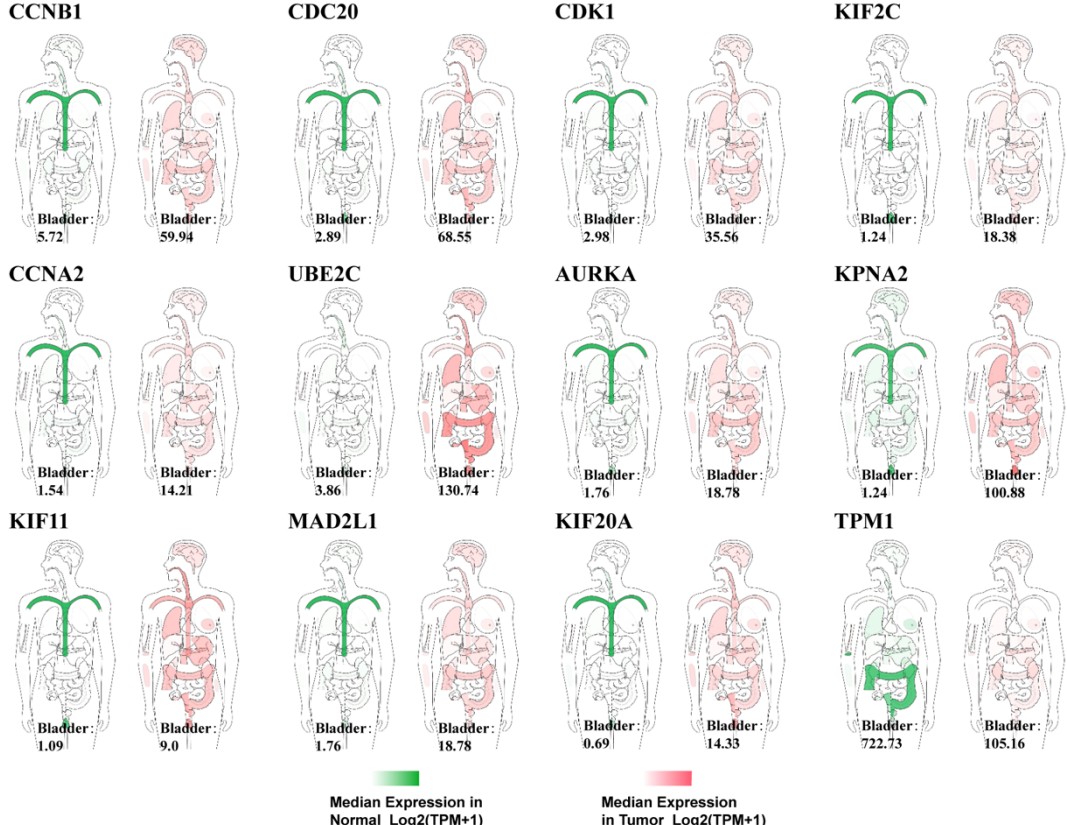

**Figure 4.** The expression body map of Hub genes. The median expressions of hub genes in tumors were marked in red and normal tissues were marked in green. The map is based on the GEPIA database (http://gepia.cancer-pku.cn), transcript per million (TPM).

## 3. Methods

The workflow of the current study is depicted in Figure 1, which has been published before [11]. For more detailed information of the datasets, materials and methods please refer to this article.

### 3.1. Data Source Identification and Data Mining

The quality control of microarray data was conducted by relative log expression (RLE) box plot through R studio (version 1-1-463). Criteria were made for defining DEGs, compared the expression levels between the non-cancerous tissues and cancer samples, where |Log FC (fold change)| > 1 and a *p*-value < 0.05 were considered statistically significant [15,40].

### 3.2. Acquisition of the Hub Genes

DEGs should at least be expressed in two different GEO datasets. The overlap between DEGs in different datasets was determined by FUNRICH software (version 3.1.3) and 418 DEGs were identified. Based on the interaction degree of 418 DEGs extracted from STRING database [14], cytoHubba analysis [18,41] reported 376 hub genes, of which 135 hub genes fulfilled the criterion of degree ≥11. However, to find the most important hub genes, we focused on the top 10 hub genes, all showing a degree ≥80. Additionally, we included another four hub genes *KPNA2*, *TPM1*, *CASQ2,* and *CRYAB*, which not only significantly correlated with overall survival based on the results from Human Protein Atlas, but also showed a degree ≥11 [15,40].

### 3.3. Functional and Clinical Analysis of the Data

FUNRICH software concerned on DEGs consisted of 11 important modules, and DAVID (Database for Annotation, Visualization and Integrated Discovery; version 6.7; http://david.ncifcrf.gov) emphasized on the 418 DEGs and 14 Hub genes [42]. Both software packages were used to annotate, visualize, and integrate the discoveries, and to extract the crucial biological information. We also used DAVID to analyze the up- and downregulated genes, separately. A Benjamini–Hochberg FDR <0.05 was considered significant in DAVID and FUNRICH analyses. In addition, we performed GO and KEGG analysis to identify the critical biological process (BP), cellular component (CC), molecular function (MP), and essential pathways potentially related to the initiation and development of BCa. $p < 0.05$ was considered statistically significant.

Clinical information was extracted from TCGA-BLCA using R software and, subsequently, the expression levels of 14 hub genes in subgroups were analyzed based on tumor stage, lymph nodal metastasis, race of patients, gender of patients, histological subtype, and molecular subtypes. To evaluate the prognostic value of the identified DEGs, we did Kaplan–Meier survival analyses of overall survival (OS) and disease-free survival (DFS). $p < 0.05$ was considered statistically significant.

### 3.4. Literature Retrieval and Oncomine Meta-Analysis

PubMed, EMBASE, Science Direct, and Google Scholar databases were used to search and identify published results about bioinformatics analysis in BCa, via the Google search engine. The data collection process was undertaken and ended in November 2019. The retrieval criteria were rigorous, filtering with bioinformatics analysis and BCa, only 7 full-text papers were returned [9,29–34]. On the basis of Oncomine database, 5 studies were relevant [35] and we gathered the information of the 14 hub genes from the 5 previous studies.

## 4. User Notes

The present report describes the character of the research "Identification of key biomarkers in bladder cancer: Evidence from bioinformatics analysis" [11]. Furthermore, the present report provides a convenient way to use extended datasets for biomarker discovery and hypothesis generation.

**Supplementary Materials:** We provide the supplementary Tables S1–S3 listing step by step the identification process of the most promising DEGs. The DEGs listed in Table S3 were used during the following analysis steps. We also indicated up- or downregulation of those DEGs (↑; ↓) in bladder cancer. Table S4 provides the 376 hub genes defined by cytoHubba. Table S5 summarizes the clinical and gene expression data of the $n = 406$ TCGA-BLCA patients for the 14 hub genes, defined from their degree of network interaction and the 11 seed genes, defined from the 11 most important modules. This table is the basis for the gene expression analyses. Tables S6–S8 summarize the results of the GO and KEGG pathway analyses. Table S9 provides the data of the Oncomine meta-analysis. The data tables may be used to construct a complete data set after applying different normalization strategies as cross-platform normalization or batch effects removal. This data set can be used as a benchmarking data set for machine learning-based feature selection in data-driven biomarker research. The following supplemental data are available online at http://www.mdpi.com/2306-5729/5/2/38/s1, Table S1: DESs extracted from different dataset, Table S2: 726 DEGs extracted from 5 GEO datasets, Table S3: 418 DEGs in BCa, Table S4: 376 Hub genes in BCa, Table S5: Clinical meta-data of 11 seed genes and 14 Hub genes, Table S6: GO and KEGG results of 418 DEGs, Table S7: GO and KEGG results of 132 upregulated DEGs, Table S8: GO and KEGG results of 286 downregulated DEGs, Table S9: Oncomine meta-analysis results of previous results.

**Author Contributions:** Conceptualization, C.Z. and J.N.; Data curation, C.Z.; Supervision, J.N.; Writing—original draft, C.Z.; Writing—review and editing, M.B.-P. and J.N. All authors have read and agreed to the published version of the manuscript.

**Funding:** This research received funding from the Leipzig University for Open Access Publishing.

**Acknowledgments:** We thank the research team of our laboratory for discussion and advice with the data analysis. We acknowledge support from Leipzig University for Open Access Publishing.

**Conflicts of Interest:** The authors declare no conflict of interest.

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
