# Peer review of "Bioinformatics Analysis Identifying Key Biomarkers in Bladder Cancer"

_data_

Round 1
Reviewer 1 Report
The authors have analyzed differentially expressed genes (DEGs) data from 6 datasets from patients with bladder cancer and controls, to identify hub genes as potential biomarkers of cancer progression by applying several bioinformatic analyses. Afterwards they contrasted the predicted “biomarkers” with the reported ones in the literature and arrived to an agreement among the biomarkers, claiming the presented data will be useful for evaluating the performance of preclinical biomarkers.
The paper is suitable for publication after addressing these comments:
- Where is the preclinical form the authors claiming is freely-available?
- Please, avoid the overuse of acronyms (GEO, TCGA-BLCA). It makes hard the reading of the paper
- Please insert, a reference for MCODE.
- Where are the GO and KEGG enrichment analyses for the 14 hub genes? (line 65-68)
- Please reference in the text the datasets used in the analyses by categories like tumor stage, lymph nodal metastasis. The expression “so on” is not scientifically elegant.
- The figure 1 must be self-explanatory and improve its quality. Figures should be understood disconnected from the text.
- In the section 2.2, authors firstly described the expression levels of 11 hub genes reported in the literature, however later they built the expression body maps for 12 hub genes…please clarify?
- I recommend to the authors connect better the experiments/analyses in order to improve the reading of the paper.
Reviewer 2 Report
The paper is a data descriptor aimed at describing data collection and analysis to support the original paper: “Identification of key biomarkers in bladder cancer: Evidence from bioinformatics analysis”. It's intended on the reusing of datasets in further research about this important topic. The proposed dataset includes new discovered hub genes in previously collected data from different sources. Broad comments The main contribution of the paper is the integration of information it presented from different data sources by using different procedures including the literature revision of previously published hub genes acting as biomarkers in bladder cancer. The results and dataset presented should be of much value for the scientist community related to this kind of disease. The authors may consider presenting a logical design of the whole dataset in order to facilitate the reusability process. They may also include a dataflow diagram to document the analytical procedure as well as an activity diagram of the data mining process including the corresponding software to be used in each step. These abstractions may be useful for readers pursuing new methods in biomarkers discovering for different diseases. Specific comments English may be improved in different lines specified in the attached document. Also some specific issues are attached. Citation of the DEGs defining criteria should be included in section 2.1. The seed definition and the procedure to obtain the 14 hub genes may be explicitly defined or cited from lines 63 to 74. Please, consider repeating Table 2 headings in the second page.

Reviewer 3 Report
The Bladder cancer topic is very relevant, so it is a very relevant element given the high incidence and percentage of recurrence of the disease. The winning of new insights into it molecular pathology and to determine potential biomarker sound well founded results. It deserve to be take in considerations.
Some consideration should be point out:
- Even when GEO (Gene Expression Omnibus), seems very familiar, it must be declared at the beginning of the work.
- I would like to know why the authors do not include in the text or supplementary material, the graphic information offered by DAVID (Database for Annotation, Visualization and Integrated Discovery ) in the annotation process.
Reviewer 4 Report
The data descriptor manuscript from Zhang et al. described briefly a pipeline that aimed at identifying key biomarkers in bladder cancer. It may be of importance for some researchers who need a quick reference of potential genes that are associated with bladder cancer. Nevertheless, the manuscript needs to be significantly improved before it could be re-considered for a publication. Please refer to the below points for more information.
- The description of the selected data is lacking. For instance, what was the type of the samples: case-control, case-adjacent control, or case-partially adjacent control. Also, other basic information regarding clinical information, analyzed platforms for transcriptome profile, etc., are needed.
- In case the study designs of included data sets are similar from one to another, I would suggest a more robust analysis pipeline with a method of meta-analysis, e.g., combined effect sizes.
- Data processing should be described in detail. It is critical for examining the quality of the subsequent analyses.
- The use of raw P-value is not appropriate for high-dimensional data analysis. A correction for multiple hypothesis testing is required, e.g., Bonferroni's adjustment of P-value cut-off or Benjamini-Hochberg procedure of False Discovery Rate as used in the section of functional analysis.
- It would be more informative to examine the survival impact using Cox regression of the selected list of "hub genes". It can be done using TCGA transcriptomics data and clinical information.
- It is quite not familiar since the concept of "hub gene" is used in the manuscript but there is no visualization of the potentially important network shown in the manuscript. I would suggest the authors make at least one example.
- Please refer to the standard nomenclature when describing genes. For instance, the italic form should be used for human genes to distinguish it with the annotation of proteins.
Round 2
Reviewer 4 Report
I thank the authors for the efforts revising the manuscript that aimed as a data description material. However, there are several points to be revised.
- The notion of P-value is not consistent in the current manuscript (P or p or p-Value, with or without italic form).
- It is best to describe potential uses of the data set. For example, some authors may apply the cross-platform normalization or batch effects removal to make one completed data set. This data set can be used as a benchmarking data set for machine learning-based feature selection in data-driven biomarker research.
- Please make sure that there are no problems with the copyright of your former paper.
Author Response
Dear Editors, dear Reviewer
Thank you very much for your helpful advice. We changed the manuscript accordingly.
Please find the point-to-point reply below. We very much appreciate your valuable time spent to help to improve our manuscript. To clearly indicate the changes made, we used the "Track Changes" function of Microsoft Word. We hope that the manuscript will now be suitable for publication in data.
Best regards,
Chuan Zhang and Jochen Neuhaus
Point-to-point response:
Reviewer 4:
I thank the authors for the efforts revising the manuscript that aimed as a data description material. However, there are several points to be revised.
Answer: Thank you very much for your kind comments.
- The notion of P-value is not consistent in the current manuscript (P or p or p-Value, with or without italic form).
We changed all occurrences of “P-value” into “p-value” and “P=” into “p=”(without italic form). Unfortunately, we were unable to change the “p-Value” in the figure 3, since it was direct output from the Oncomine software.
We updated figure 3 accordingly.
- It is best to describe potential uses of the data set. For example, some authors may apply the cross-platform normalization or batch effects removal to make one completed data set. This data set can be used as a benchmarking data set for machine learning-based feature selection in data-driven biomarker research.
Thank you for this suggestion. We want to ask for your permission to integrate your last sentence into our explanation. We could not think of a more appropriate phrase. In case you do not agree, please let us know. Thank you!
We included the following description of the supplemental tables into the section
Supplementary Materials:
We provide the supplementary tables S1-S3 listing step by step the identification process of the most promising DEGs. The DEGs listed in table S3 were used during the following analysis steps. We also indicated up- or downregulation of those DEGs (; ¯) in bladder cancer. Table S4 provides the 376 hub genes defined by cytoHubba. Table S5 summarizes the clinical and gene expression data of the n=406 TCGA-BLCA patients for the 14 hub genes, defined from their degree of network interaction and the 11 seed genes, defined from the 11 most important modules. This table is the basis for the gene expression analyses. Tables S6-8 summarize the results of the GO and KEGG pathway analyses. Table S9 provides the data of the Oncomine meta-analysis.
The data tables may be used to construct a complete data set after applying different normalization strategies, e.g. cross-platform normalization or batch effects removal. This data set can be used as a benchmarking data set for machine learning-based feature selection in data-driven biomarker research.
- Please make sure that there are no problems with the copyright of your former paper.
Thank you for this advice. The diagnosis paper is an open access publication without any copyright restrictions. We ensured proper citation of the used figures.